# GWLBC: Gray Wolf Optimization Based Load Balanced Clustering for Sustainable WSNs in Smart City Environment

**DOI:** 10.3390/s22197113

**Published:** 2022-09-20

**Authors:** Surjit Singh, Srete Nikolovski, Prasun Chakrabarti

**Affiliations:** 1Computer Science and Engineering Department, Thapar Institute of Engineering & Technology, Patiala 147004, India; 2Power Engineering Department, Faculty of Electrical Engineering Computing and Information Technology, 31000 Osijek, Croatia; 3School of Computer Science Engineering and Technology, ITM SLS Baroda University, Vadodara 395150, India

**Keywords:** improved gray wolf optimization, clustering, load balancing, performance modeling, sustainable WSNs

## Abstract

In a smart city environment, with increased demand for energy efficiency, information exchange and communication through wireless sensor networks (WSNs) plays an important role. In WSNs, the sensors are usually operating in clusters, and they are allowed to restructure for effective communication over a large area and for a long time. In this scenario, load-balanced clustering is the cost-effective means of improving the system performance. Although clustering is a discrete problem, the computational intelligence techniques are more suitable for load balancing and minimizing energy consumption with different operating constraints. The literature reveals that the swarm intelligence-inspired computational approaches give excellent results among population-based meta-heuristic approaches because of their more remarkable exploration ability. Conversely, in this work, load-balanced clustering for sustainable WSNs is presented using improved gray wolf optimization (IGWO). In a smart city environment, the significant parameters of energy-efficient load-balanced clustering involve the network lifetime, dead cluster heads, dead gateways, dead sensor nodes, and energy consumption while ensuring information exchange and communication among the sensors and cluster heads. Therefore, based on the above parameters, the proposed IGWO is compared with the existing GWO and several other techniques. Moreover, the convergence characteristics of the proposed algorithm are demonstrated for an extensive network in a smart city environment, which consists of 500 sensors and 50 cluster heads deployed in an area of 500 × 500 m^2^, and it was found to be significantly improved.

## 1. Introduction

Presently, the research is condensed toward computational intelligence, energy efficiency, and the smart sensor’s design and deployment. In the modern world of smart communication, computation, and operation, computational intelligence has revolutionized research in every sphere of engineering. In a smart city environment, quick and appropriate action needs to be taken with no time. This requires the effective communication and the deployment of smart sensors for reliable information exchange among sensors and cluster heads and hence to develop a sustainable WSN. The coverage of these networks will be broad for a long duration, which leads to customers’ satisfaction and the reliability of system operation in a smart city environment. In the past, several computational techniques have been developed for the load-balanced clustering of WSNs. Usually, their performance is explored based on their computational capability and the improvement in the objective functions in terms of the network lifetime, energy consumption and coverage area.

For quick and reliable operation, it is believed that the swarm intelligence approaches are found to give better results than the evolutionary approaches. The variation in the objective functions mainly depends upon the constraints and the operator experience while updating every new fitness function and the problem initialization. Recently, a nature-inspired gray wolf optimization (GWO) approach was developed in [1]. The authors have explored the performance of GWO compared to the several existing computational methods, and it is found to be encouraging. The concept is derived from the fact that wolves live in a group of five to twelve members, and depending upon their skill, the role of each member is different. Therefore, the simulation performed in this optimization technique is based on the leadership hierarchy of gray wolves and their hunting behavior. In GWO, wolves are categorized into four classes, namely Alpha, Beta, Delta and Omega. In this hierarchy, Alpha is represented as the group leader, and Beta is the subordinate of the group leader. Conversely, Delta is the subordinate to both Alpha and Beta, whereas Omega are the other wolves that serve as soldiers in the group.

The optimization function in GWO is derived from the three phases of hunting: searching the prey, encircling the prey, and attacking the prey. In the proposed work, the GWO technique is explored to obtain the load balancing of the gateways in heterogeneous WSNs, since clustering is a discrete problem and the fitness function can be improved by tweaking the existing GWO approach for different objectives with different operating constraints. In this work, an improved GWO is proposed for the sustainable WSNs in a smart city environment where the performance matrix involves the network lifetime, coverage area and energy consumption during information exchange among sensors and cluster heads for the WSN under consideration.

### 1.1. Related Work

For sustainable cities, the deployment of sensors may be employed for several purposes such as smart energy management for electricity, water and gas, climate change due to temperature, humidity and air quality, smart governance for economy and trade, smart transportation and traffic control, and the smart education system. Smart sensors are usually operating wirelessly, and these wireless sensor networks are built with many sensors deployed in inaccessible locations [2]. These sensors’ size is minimal and has limited computational and processing resources [3,4]. In addition, during the mobility of sensor nodes, re-authenticating and locating the node movement is a security issue in WSNs [5]. The sensors are deployed at several locations and operate in clusters. These clusters have their cluster heads [4] and work as gateways that aggregate the information and forward it to the base station [6]. In practice, the sensors assigned to a particular cluster can be reassigned to another cluster head depending on the type and priority of information exchanged during operation [7]. Here, it can be noticed that clustering is a discrete problem, and hence, it is required to develop some heuristic or metaheuristic approaches to achieve an energy-efficient load-balanced clustering of WSNs.

In a smart city environment, the performance of the sustainable WSNs needs to be analyzed in terms of the network lifetime, coverage area, and energy consumption during information exchange. In the literature, the researcher has explored several nature-inspired approaches with different objectives under different operating conditions and constraints. The authors in [8,9,10] have presented the harmony search algorithm, genetic algorithm and the least-distance-based load-balanced clustering of WSNs for the objective of the number of active sensor nodes, cluster heads, and convergence rate.

Similarly, a heuristic approach has also appeared in [11], where clustering is performed based on lingering and residual energy. However, this approach is suitable for small networks. The authors in [12] divided the network geographically into two parts, and the cells are allowed to communicate in two categories where the operator predefines the operating rules. Hence, it may not yield a good solution under the change in operating constraints for different objectives. In [13], the authors have presented fog computing for the efficient clustering of large WSNs, and in [14], a fuzzy C-means algorithm has appeared for the objective of active sensors and scalability. In [15], a routing algorithm is developed for IoT-based modules to reduce end-to-end delay and an improved lifetime of nodes in a wireless mesh network. Talaat et al. [16] realized the application of smart sensors in a smart grid application for energy management.

Several other computational techniques, such as genetic algorithm (GA) in [17], simulated annealing (SA) in [18], particle swarm optimization (PSO) in [19], harmony search algorithm (HSA) in [20,21,22], ant colony optimization (ACO) in [23], artificial bee colony (ABC) in [24], and gravitational search algorithm (GSA) in [25], have also appeared in the literature. Here, the PSO is good at small and medium-size problems, while HSA is good at big-size problems, because PSO has prematurity due to global-best-oriented searching [26]. However, the authors in [27] presented big data grouping into a cluster and proposed automated clustering using hybridized PSO. In these works, the authors’ motive is to compare the convergence characteristics of their approaches. In contrast, a performance matrix needs to be developed to evaluate the WSNs for sustainable smart cities comprehensively. The authors in [28,29,30] conferred that the wireless sensor networks are elemental to build smart systems in smart cities and proposed IoT-based infrastructures for effective communication among sensors. However, the deployment of these sensors and mutual information exchange is another important issue to be explored for reliable operation for customers’ satisfaction.

Furthermore, a recently developed GWO has been demonstrated for 29 well-known objective functions, and the experimental results show that GWO outperformed the existing computational intelligence techniques, such as, GSA, PSO, HSA, differential evolution (DE) and evolutionary programming (EP) [1]. Recent trends reveal that the intelligent computational methods are usually adopted for multi-objective discrete problems such as clustering in WSNs [31,32,33]. The dual gray wolf optimization (GWO) algorithm has been refined, and binary and dogmatic components have been proposed in [34]. However, it has been observed that out of 29 objectives, GWO is not used for the load-balanced clustering of WSNs.

Researchers, manufacturers, and application developers have recently shown a lot of interest in the development of new digital industrial technology known as Industry 4.0 [35,36]. Industry 4.0 aims to connect and integrate all elements of the conventional manufacturing environment in order to facilitate quicker, more adaptable, and more efficient production processes that result in higher-quality products at lower costs. For use in smart grid applications, the authors in [37] suggested a unique dynamic clustering-based energy-efficient and QoS-aware routing protocol that was inspired by several aspects of bird mating optimization (BMO). In both sparse and dense network deployments, the suggested distributed approach increases network stability and reduces wasteful packet retransmissions [37]. A multi-objective tournament harmony search-based load-balanced clustering technique was presented by Singh and Kumar [38]. By changing the status of the on-duty sensors to off-duty and the off-duty sensors to on-duty, respectively, the coverage conscious memory consideration, random selection, and tournament selection-based pitch modification to eliminate clumps and filling voids is enforced during harmony improvisation.

In addition, an AI-based drone routing strategy is modified for data collection to get around the hot spot issue in [39]. The suggested method is contrasted with currently used clustering techniques for fixed WSNs, such as LEACH, LEACH-C, and LEACH-B [39]. The authors of [40] presented a unique routing system for wireless ad hoc and sensor networks for supervised device data transfer from smart grid generators to the command-and-control center. Due to each generator’s complex mechanical design, the protocol expects that numerous sensor devices, i.e., temperature sensors, oil level sensors, turbine status sensors, etc., will be installed.

In order to study and extract network behaviors in cellular networks for Industry 4.0 applications from a big data perspective, Jiang et al. [41] presented a big data-based analytical framework employing Hadoop, Hive, and HBase. To analyze the performance of long-lived C-TCP flows across Industry 4.0 WiFi infrastructure while accounting for all losses, Pokhrel and Singh [42] created a thorough model. In order to investigate the effects of WiFi system parameters on transport-layer performance and fairness, the proposed mathematical model incorporates WiFi system factors including the retransmissions cap and the AP buffer size.

Industrial wireless sensor networks have more stringent standards for energy homogeneity, real-time data transmission, and energy consumption. In order to meet these requirements, a novel energy-efficient clustering method known as quantum elite gray wolf optimization, which is inspired by quantum-related bio-inspired optimization, is suggested in [43].

According to the pertinent research, Industry 4.0 load-balanced clustering can indeed be accomplished using advanced optimization approaches. The proposed work, however, offers an enhanced GWO for load-balanced clustering that is energy-efficient. In order to construct sustainable WSNs for sustainable cities, the results are robustly compared with a number of different ways in terms of the performance matrix of dead CHs and sensors, network longevity, and energy usage.

### 1.2. Motivations

The literature reveals that the performance of WSNs depends upon several parameters in a smart city environment. The population-inspired approaches are more persuasive than single-solution approaches, and they have some advantages, which are listed below:

The results may be obtained in sudden jumps toward the targeted part of the solution space.Local optima are avoided in the case of meta-heuristics. Multiple solutions force each other to obtain the best result, improving the reliability during operation.These approaches have more remarkable exploration ability in comparison with single solution approaches.Swarm intelligence is playing excellently among these meta-heuristics. Swarm intelligence approaches have some significant advantages over others:⚬These approaches store the previous information about the search space in different iterations, whereas population-inspired algorithms discard the knowledge of the last solutions.⚬These algorithms use memory to preserve the best solution, and they are easy to implement for fast and accurate results.⚬These algorithms, in general, have few parameter settings and fewer operators compared to population-inspired approaches, and hence, the convergence ability is fast.

Researchers have proposed several swarm intelligence algorithms for clustering in WSNs to achieve best fitness and better convergence rate. In this scenario, the computation process needs to be improved in the exploration and exploitation stage of a swarm-inspired technique to find the best solution by considering global search space. Considering the above fact, in this work, the computational ability of gray wolf optimization is explored for the load-balanced clustering in WSNs in a smart city environment. The advantages of GWO are:It requires significantly less storage.The convergence is faster due to the continuous reduction in each search agent.

### 1.3. Proposed Work

The contributions of the proposed work are as follows:Several performance parameters are represented with improved objective function, and load-balanced clustering is performed using traditional GWO and an improved GWO.For fast convergence, the hunting mechanisms in the GWO hierarchy are improved.The convergence rate of the proposed technique i.e., improved gray wolf optimization (IGWO) based load-balanced clustering, is explored compared to traditional GWO and HSCA [8] for fitness function.

Moreover, the network performance analysis is also demonstrated for HSCA [8], GALBCA [10], and traditional GWO compared to the proposed IGWO.

## 2. Problem Formulations

The heterogeneous WSN model is considered and, in this network, the disjoint sets of sensors form a cluster. Each cluster has a gateway associated with it to send the data packets received from the sensor nodes. Later, the information is aggregated, and it is communicated to the base station [6]. In this process, the power consumption of the receiver and transmitter leads to the energy dissipation of the radio due to free space and the multi-path fading channel [44]. Therefore, the lifetime of the networking is majorly affected by the power consumption, which demands load balancing among the cluster heads, i.e., gateways. With symbols having their usual meanings [8,44], the problem of the load balancing of gateways is formulated for minimizing the load of the maximum loaded gateway, which is as follows:(1)$$Minimize\;G_{TL}=\max\{G_{L}|where,\;G_{j}\in G\;\}$$

The above problem is subject to the following constraints,

(a)**Threshold distance:** The threshold distance of sensors from the CH is determined and used to generate the initial solution space. Here, *Max_G* is the maximum loaded CH with *p* number of sensors connected and *Min_G* is the minimum loaded CH associated with *q* number of sensor nodes. *S* is the total set of *p* and *q*, and the sensor connected to these CHs is the union of both groups as it is given in (2),


(2)
$$S\in \left\{\overset{p+q}{\underset{k=1}{\cup }}S_{k}\right\}$$



(3)
$$\sum_{s_{j}\in S}\mathrm{dist}\left(i,j\right)\;\leq t\_\mathrm{dist}\;|\;where,\;G_{i}\in G_{s}$$


In Equation (3), t_dist is the distance threshold of the sensor from each CH, and it is calculated as under,
(4)$$t\_dist=\frac{1}{S}\sum_{i=1}^{S}\sum_{j=1}^{2}dist\left(i,j\right)\times b_{ji}\;\;$$

In Equation (4), *S* is the number of sensors in the maximum and minimum loaded CH set. *b_ji_* is used for the connection between the sensors and CHs.

(b)**Disjoint set:** The disjoint sets allow each sensor node to connect with only one CH.


(5)
$$\sum_{G_{i}\in G_{s}}b_{ji}=1\;|\;where,\;s_{j}\in S\;$$


In Equation (5), if *b_ji_* is 1, then it is the member of the disjoint set; otherwise, it is not a part of it.

(c)**Gateway load:** The traffic load of a CH *G_i_* should be less than or equal to the maximum load *G_TL_* of a maximum loaded CH.


(6)
$$\sum_{s_{j}\in S}d_{j}\;.\;b_{ji}\leq G_{TL}\;|\;where,\;G_{i}\in G_{s}$$


In Equation (6), *d_j_* is the load of sensor *s_j_*, and *s_j_* belongs to the disjoint set of *G_i_.*

## 3. Gray Wolf Optimization

The GWO is a swarm intelligence-inspired computational approach stimulated by the deeds of gray wolves [1]. The swarm intelligence-inspired optimization techniques are competent in finding the best solution. The procedural steps of GWO are described in the next part of this section.

Gray wolves belong to Canidae ancestors, and they are at the peak level of the food procession. Generally, they always exist in a pack with a set of 5–12 members and follow a strict societal ladder as depicted in Figure 1. The simulation modeling of GWO is based on the leadership ladder of gray wolves and their hunt manners. These are categorized into three levels and four types, which are described below.

### 3.1. First Level

***Alpha****: Leader:* In the leadership hierarchy, leaders at the top level are called Alpha. They are decision-makers, and while hunting, they decide where to rest, when to move, etc. The leader wolf can be a male or female. They are best at finding prey locations, whereas they may not be the most muscular wolf. The group strictly follows the leader’s decision. The other wolves follow democratic behavior and act the same in the group. All group members acknowledge the leader by holding their tails down. In every situation, all members should follow the order of the leader wolf. This wolf is also known as a dominant wolf. They are responsible for managing the pack and maintaining discipline in the group.

### 3.2. Second Level

***Beta****: Subordinate to leader:* The wolves that bring the leader to power through fitness influence are called Beta. These wolves are subordinate to the Alpha and take responsibility for the group in their absence. The Beta wolf works as a consultant to the Alpha wolf and helps in the decision of Alpha for group activities. Similar to the Alpha, they can also be a male or female. Therefore, the Beta wolf is next to Alpha in the leadership hierarchy and suitable for holding the leading position when required. After the retirement of Alpha, these wolves are the natural successor to them. However, this wolf should respect the leader and instruct lower levels in the hierarchy. Therefore, this wolf plays the role of advisor, reinforces the leader’s instructions throughout the group, and provides feedback to the leader.

***Delta****: Subordinate to leader:* These wolves are also subordinate to the Alpha wolf and act as devotees. These wolves have to support Alpha and Beta together. They dominate Omega wolves and work as caretakers, hunters, elders and sentinels. They take care of the weak, ill, etc., wolves in the group. If a wolf is not in the category of Alpha, Beta, and Omega, it is a Delta wolf. These wolves are also responsible for watching boundaries, the safety of the group, and warning the group.

### 3.3. Third Level

***Omega****: Other wolves or soldiers:* In the hierarchy, they work as a soldier and work by alerting the group and assisting the Delta wolves in protecting the boundaries. These wolves acquire food last and provide food to the other dominants. It seems that these wolves are not very important for the group. However, the whole pack faces problems in the absence of Omegas because they assist in satisfying the entire group. However, the entire group may face internal problems due to frustration regarding a lack of Omegas. Sometimes, Omegas are babysitters in the group.

Based on the above three-level hierarchy, this work proposes the gray wolf optimization (GWO) clustering algorithm for WSNs. The GWO-based clustering algorithm can select the minimum loaded CH successfully and minimize the load of maximum loaded CHs.

In addition to the above, the method of hunting is an interesting behavior of wolves. According to Mirjalili et al. in [1], the steps followed by wolves during hunting are given below:Track, chase and approach the food.Pursue, surround, and irritate the prey until it stops moving.Hit in the direction of the prey.

The following section presents a load-balancing algorithm based on hunting and the societal ladder of gray wolves to minimize the load of maximum loaded CH.

## 4. Improved GWO-Based Load-Balanced Clustering

This section presents the proposed load-balanced clustering based on GWO and the illustrations of the societal ladder, initialization, tracking, encircling, and attacking prey.

### 4.1. Illustration of the Social Hierarchy in Proposed Load-Balanced Clustering Problem

The societal ladder of gray wolves in GWO is modeled as under:❖ *The fittest solution is called the **Alpha (α)**.*❖ *The following subsequent solutions are called **Beta (β)** and **Delta (δ)**.*❖ *The **Omega (ω)** wolves represent the rest of the candidate solutions. These wolves are the followers of the above three wolves.*

The resulting behavior of this approach depends upon the three wolves, i.e., *α*, *β*, and *δ.* The social hierarchy to model the gray wolf optimizer for load balancing in heterogeneous WSNs is depicted in Figure 2.

Algorithms 1 and 2 show the pseudo-code of traditional gray wolf optimization (GWO-based load-balanced clustering algorithm and the proposed load-balanced clustering algorithms (IGWO).
**Algorithm 1:** Pseudo-code of traditional gray wolf optimization (GWO)-based load-balanced clustering algorithm**Step1**: Initialize the population of gray wolves ***G_i_*** (*i =* 1, 2, …, *n*)**Step2**: Initialize *a, A, C***Step3**: Calculate the fitness of each agent in search space**Step4**: ***G_alpha_*** = the fittest wolf**Step5**: ***G_beta_*** = the second-best gray wolf**Step6**: ***G_delta_*** = the third-best gray wolf**Step7**: Repeat ***while itr* < *Round*****Step7.1**: ***for*** each ***i*th** sensor in each best search agent gray wolf ***do*****Step7.1.1:** Find ***i*th** maximum loaded CH ***Max*** and minimum loaded CH ***Min*** in ***Alpha****, **Beta*** and ***Delta***.**Step7.1.2:** Perform random load-balanced clustering for all best search agents ***A**lpha***, ***Beta*** and ***Delta*** resp.    **Step7.1.2.1**:***If A <***
**1**    ***[Search into Initial Solution Space* i.e., *Exploitation]***    ***[//Call Exploitation();]***    //Follow random approach to perform load-balanced clustering.    **Step7.1.2.1.1**: Set ***G_Load*1 = *Galpha[i]*.**    **Step7.1.2.1.2**: Set ***Galpha[i]* = *Gbeta[i]*.**    **Step 7.1.2.1.3**: Set ***Gbeta[i]* = *G_Load*1**     *//Similarly update delta and omegas*    **Step7.1.2.2**: *Else Diverge or Perform Exploration*    ***//Look in to global space***    ***//Call Exploration();***    **Step7.1.2.2.1:** Initialize ***G*** from the global search space.    **Step7.1.2.2.2:** Replace ***Galpha[i]*** with ***G***.    ***[End of if]***   ***[End for loop]*****Step7.2**: Update ***a****, **A**, **C*****Step7.3**: Calculate the fitness of each search agent.**Step7.4**: Update ***G_alpha_***,***G_beta_***,***G_delta_*****Step7.5**: Increment ***itr******[End while loop]*****Step8**: Return best gray wolf ***G_alpha_***

**Algorithm 2:** Pseudo-code for the proposed approach**Step1**: Initialize the population of gray wolves ***G_i_*** (*i* = 1, 2, …, *n*)**Step2**: Initialize ***a****, **A**, **C*****Step3**: Calculate the fitness of each agent in search space**Step4**: ***G_alpha_*** = fittest wolf**Step5**: ***G_beta_*** = second best wolf **Step6**: ***G_delta_*** = third best gray wolf **Step7**: Repeat ***while itr* < Round, then *do*****Step7.1**: ***for*** each gray_wolf ***do*****Step7.2**: ***for*** each ***i*th** sensor in each best search agent gray wolf ***do*****Step7.2.1**: Find ***i*th** maximum loaded CH ***Max*** and minimum loaded CH ***Min*** in ***Alpha****, **Beta*** and ***Delta***. **Step7.2.2**: Perform load-balanced clustering for all the best search agents ***alpha***, ***beta*** and ***Delta***, respectively.**Step7.2.3**: ***If A <* 1**    ***[Search into Initial Solution Space* i.e., *Exploitation]***    ***[//Call Exploitation();]***    //Update ***Galpha***    **Step7.2.3.1:** Set ***G_Load*1 = *Gbeta[i]***    **Step7.2.3.2:** Set ***G_Load*2 = *Gdelta[i]*.**         ***If G_Load*1*< Galpha[i] && G_Load*1 *< G_Load*2**           ***Galpha[i] = G_Load*1**         ***Else if G_Load*2 *< Galpha[i]***           ***Galpha[i] = G_Load*2**         ***[End]***    **Step7.2.3.3:** Set ***G_Load*1 = *Gbeta[i]*.**    // Update ***Gbeta***    **Step7.2.3.4:** Set ***G_Load*2 = *Gdelta[i]*.**         ***If G_Load*1 *> G_Load*2**           ***Gbeta[i] = G_Load*2**         ***Else***           ***Gbeta[i] = G_Load*1**         ***[End]***     *//Similarly update delta***Step7.2.4**: *Else diverge or perform exploration*    ***//Look in to global space***    ***//Call Exploration();***    **Step7.2.4.1:** Initialize ***G*** from the global search space.    //Update ***Galpha***    **Step7.2.4.2:** Set ***G_Load*1** = ***Galpha[i]***.           ***If G_Load*1 *> G***                ***Galpha[i] = G***           ***Else***
                ***[No change]***           ***[End]***    //Update ***Gbeta***    **Step7.2.4.3:** Set ***G_Load*2** = ***Gbeta[i]***.           ***If G_Load*1 *> G***                ***Gbeta[i] = G***           ***Else***
                ***[No change]***           ***[End]***    *//Similarly update delta****[End of if in* Step7.2.3*]***[End for loop]**Step7.3**: Update ***a****, **A**, **C*****Step7.4**: Calculate the ***fitness*** of each gray wolf **Step7.5*:*** Update ***G_alpha_***, ***G_beta_***, ***G_delta_*****Step7.6**: Increment ***itr******[End while loop]*****Step8:** Return best gray wolf ***G_alpha_***

### 4.2. Illustration of Encircling Prey

The following equations represent the mathematical model of encircling prey during hunting:(7)D=|CXp−A·X(t)|
(8)X(t+1)=|Xp(t)−A·D|

*A* and *C* are coefficients.
(9)A=2·a·r1−a
(10)C=2·r2 
where


*t is current iteration*


*X_p_(t) load of CH in t*th *iteration*

*A decrease linearly from* 2 *to* 0

*r*1 *and r*2 *are random values chosen between* 0 *and* 1.

### 4.3. Illustration of Initialization of Initial Solution

The initial solution is generated based on the threshold value of the distance between the sensors and CHs. The threshold distance is defined in the problem formulation of the proposed approach to create disjoint sets of sensors in the initial solution space. However, the traditional GWO initializes the solution space randomly.

**Illustration** **1:**
*Figure 3 shows the initial solutions where Alpha, Beta, Delta and Omega represent the societal ladder of gray wolves based on their fitness value. In the first vector **Alpha**, sensor 1 is associated with CH number 46, sensor 2 is associated with CH 22, etc. In the second vector **Beta**, sensor 1 is associated with CH number 4, sensor 2 is associated with CH 44, etc. Similarly, in the **Delta**, sensor 1 is associated with CH number 36, sensor 2 is associated with CH number 19, and so on. As shown in Figure 3, the other search agents are Omegas, and they are represented similarly to Alpha, Beta and Delta.*


**Remark** **1:**
*As shown in Figure 3, for the initial solution space, the CHs are picked out based on the least distance from a threshold value from the list of all possible CHs. However, the **Alpha, Beta, Delta** and **Omegas** are generated according to their fitness values in the population matrix.*


The fittest solution is represented with Alpha. The second and third best solutions are represented with ***Beta*** and ***Delta***, whereas others are regarded as the ***Omegas***. This process allows increasing the network lifetime by reducing energy consumption compared to traditional gray wolf optimization (GWO) due to less distance between the sensor and CH and the social hierarchy of GWO.

Generate 500 populations using Algorithm 3: Create a population matrix M2 12 × 500 with a fitness values column (*f*(*x*)) added. As a result, the matrix’s overall dimensions are 12 × 501.


(11)
$$M=\left[\begin{array}{ccccc} x_{1}^{1} & x_{2}^{1} & \ldots & x_{500}^{1} & f(x^{1})\\ x_{1}^{2} & x_{2}^{2} & \ldots & x_{500}^{1} & f(x^{2})\\ \vdots & \vdots & \vdots & \vdots & \vdots \\ x_{1}^{12} & x_{2}^{12} & \ldots & x_{500}^{12} & f\left(x^{12}\right) \end{array}\right]$$


**Algorithm 3:** Pseudo-code to generate population M2 matrix*for I =* 1:12   *For j =* 1:500      *//Select CH from list of possible CHs between* 1 *and* 50   *X[i][j] = rand (*1,50*);*   *[End for loop]**Evaluate Fitness(X[i][j]);**F[i]= Fitness(X[i][j]);**[End for loop]**---------------------------------------------------------------------------------------------------------------------------****Where****,*0 *< X_i,j_ < =* 50

### 4.4. Illustration of Hunting Behavior

The leader *A**lpha* guides hunting, but *Beta* and *Delta* also participate in the hunting. These wolves are capable of encircling and originating the food location. Initially, in the search space, there is no idea about the best solution (prey location). Therefore, to find the best solution or the place of prey, it is to be considered that the *Alpha,* i.e., best fittest solution, *B**eta,* i.e., second best, and *Delta,* i.e., third-best have better knowledge about the food location. As shown in Figure 3, the best three solutions, *Alpha, Beta* and *Delta*, are stored. The *Omegas* follow the order of these three best search agents and update their positions accordingly.
(12)Dalpha=|C1Xalpha−X|
(13)Dbeta=|C1Xbeta−X|
(14)Ddelta=|C1Xdelta−X| 
(15)X1=|Xalpha−A1·Dalpha|
(16)X2=|Xbeta−A2·Dbeta| 
(17)X3=|Xdelta−A3·Ddelta|

The CH load update equation is:(18)$$X\left(t+1\right)=\frac{X1+X2+X3}{3}\;$$

Here, a vector is used for exploration (searching) and exploitation (attacking).
(19)$$a=2-t\cdot \frac{2}{\max\_itr}\;$$

Figure 4 shows that a maximum loaded CH load is minimized according to *Alpha, Beta*, and *Delta* in the search space. The final solution would depend upon the three *Alpha, Beta* and *Delta* search agents. Conversely, *Alpha, Beta* and *Delta* estimate the best solution, and other candidate solutions are updated randomly.

**Illustration** **2:**
*Figure 4 shows that the CH number 46 in the **Alpha** vector at the first location has a maximum load compared with the **B****eta** CH number 4 at its first location (if matched in possible CHs list of sensor #1). Similarly, CH number 22 in the **Alpha** vector at the second location has a maximum load compared with the B**eta** CH number 44 at its second location (if matched in possible CHs list of sensor #2). In the case of sensor #5, the minimum loaded CH was replaced with **Delta** solution vector CH 60 (if matched in possible CHs list of sensor #5). Similarly, **Beta** and **Delta** wolves are updated during this phase.*


**Remark** **2:**
*The convergence of IGWO is fast compared to the traditional GWO due to the substitute of maximum loaded CH in **Alpha** vector from **Beta** and **Delta**. This part of the IGWO replaces the less loaded CH from **Beta** and **Delta** with the maximum loaded CH of Alpha, resulting in faster convergence.*


**Lemma** **1:**
*The new best search agents (**Alpha, Beta** and **Delta**) formed in the above hunting stage are legitimate.*


**Proof:** The ***Alpha***, ***Beta***, ***Delta*** and ***Omegas*** are generated according to their fitness values in the population matrix. It guarantees that CH chosen from the subsequent gray wolf is less loaded. Therefore, the new agent formed during this stage ensures its validity by obtaining better fitness. □

### 4.5. Illustration of Attacking Prey

The gray wolf stops hunting when the prey stops moving. The value of ‘*a*’ decreased to approach the prey. The value of ‘*A*’ depends on ‘*a*’. Here, ‘*A*’ is a random value in between the interval [−2a, 2a], and ‘*a*’ is decreased from 2 to 0 in each round of iteration. If the random value of ‘A’ is in the range [−1, 1], the next location is between its current location and prey. If |A| < 1, then the wolves attack the prey, as shown in Figure 5. This reinforces taking the next value from the initial solution space; therefore, the next minimum loaded CH is chosen from the initial gray wolf. This is because with the use of *alpha*, *beta* and *delta* operators, the GWO algorithm might be stuck in local solutions. To overcome this problem, it is required to introduce more operators and greater exploration of the solution space.

During this phase of traditional GWO again, a random selection has been performed. In the case of the IGWO, this phase works similarly to the previous phase. A replacement is performed when a gray wolf attacks the prey from the initial solution with less loaded CH; otherwise, it is discarded.

**Remark** **3:**
*In the exploitation mechanism of IGWO, the convergence rate will always be faster with replacement of maximum loaded CH in **Alpha, Beta** and **Delta** solution vectors from the initial solution space. Similar to hunting, this phase of the proposed approach guarantees the fast convergence of the solution.*


**Lemma** **2:**
*The new agent created in the **exploitation** phase is valid.*


**Proof:** The ***Alpha***, ***Beta***, ***Delta***, and ***Omegas*** agents update their values only when the less loaded gateway is selected from the subsequent agent. Therefore, the new agent created during exploitation is valid. The resulting agent will always be better than the previous agent. □

### 4.6. Illustration of the Search for Prey (Exploration)

This section presents the exploration of solution space in GWO. Gray wolves generally search to follow the ***Alpha***, ***Beta*** and ***Delta*** wolves. The solution may exist in the global search space. Therefore, A’s value is used to diverge from the solution if the value of A is greater than 1 or less than −1. Due to this, it can explore the global search space. In this phase, the less loaded CH is selected to update; otherwise, it is discarded. It diverges from prey to hopefully find the fittest prey, as shown in Figure 6, considering that a better solution may exist in the global search space. Therefore, this phase tends to go for it by avoiding local optima.

**Remark** **4:**
*The convergence of the proposed exploration approach is faster compared to the traditional GWO exploration phase. In the traditional GWO exploration phase, the convergence becomes sluggish due to random selection. Higher loaded CH from the global search space also has the chance of replacement with less loaded in the solution vector. This may result in divergence. Therefore, the proposed approach for exploration will always converge toward the optimum value of the fitness function and result in faster convergence.*


**Lemma** **3:**
*The new solution vector produced in the above exploration process is valid.*


**Proof:** At the time of exploration, a less loaded CH is selected from global search space to update; otherwise, it is discarded in this phase. Therefore, it always replaces a less loaded CH from the global search space with a maximum loaded CH in the solution vector. Hence, the exploration process does not hamper the validity of the solution vector. □

Primary considerations of GWO for energy-efficient load-balanced clustering are described as under the following:The social hierarchy of ***Alpha***, ***Beta***, ***Delta*** and ***Omega*** wolves is proposed for load-balanced clustering to determine the optimal solution.The proposed encircle behavior minimizes a load of maximum loaded CH. It is considered that the best search agents ***Alpha***, ***Beta,*** and ***Delta*** maintain the knowledge of the desired solution.The proposed hunting mechanism of the load-balanced clustering algorithm reinforces the best search agents to find the best solution.The values of ***a*** and ***A*** help with exploring the solution in the global search space and prevent stagnation from local optima.These values of ***a*** and ***A*** smoothly balance the load of CHs by transitioning in between exploration (searching) and exploitation (attacking).Therefore, this algorithm has crucial parameter settings that need to be adjusted.

## 5. Test System, Assumptions and Result Analysis

In a sustainable cities environment, the proposed algorithm is demonstrated for a large network that consists of 500 sensors and 50 CHs deployed in an area of 500 × 500 m^2^. The various parameters for performance analysis of the WSN parameters are depicted in Table 1. The simulation is performed in the MATLAB environment, and the results are obtained using IGWO, traditional GWO, GALBCA and HSCA. The GWO-based approaches are considered as traditional GWO and improved-GWO (IGWO) based load balanced clustering.

Furthermore, the performance of IGWO and traditional GWO is evaluated. The results obtained using the proposed approach are compared with several other recently developed computational approaches: for example, improved harmony search and evolutionary approach, which are available in the related literature. The performance metrics used to assess the network performance includes dead CHs, dead sensors, energy consumption and network lifetime. The result analysis of proposed load-balanced clustering is demonstrated for convergence ability and performance matrix in a smart city environment.

### 5.1. Convergence Characteristics

The convergence rate is evaluated for IGWO and HSCA by considering the standard deviation. The lower value of standard deviation represents the better fitness value.

Table 2 shows the variation in standard deviation for IGWO and HSCA. Here, the standard deviation is evaluated for 500, 1000 and 1500 rounds, and the results are represented in Figure 7, Figure 8 and Figure 9, respectively. The performance of HSCA and IGWO is compared. Figure 7 shows the variation in standard deviation for HSCA and IGWO. Here, it is noticed that the performance of the IGWO is better compared to the HSA. For HSA, the standard deviation is found to be minimum in 450 rounds, whereas this value is minimum even in less than 400 rounds in case of IGWO. Conversely, comparing the standard deviation for these two approaches, the case of IGWO is found to be low in the different number of rounds.

Figure 8 and Figure 9 show that the value of standard deviation, i.e., 2.2516, is achieved for IGWO in less than 1000 rounds, whereas in Figure 9, HSCA represents the value of standard deviation, i.e., 2.4651, in more than 1300 rounds. In this case, as well, the proposed approach converges in less than 1000 rounds only. Here, it can also be observed that the proposed IGWO-based efficient load-balanced clustering is more effective than the HSCA.

### 5.2. Performance Evaluation for Sustainable WSNs

For sustainable cities, the performance of WSNs needs to be evaluated in terms of several parameters such as dead sensor nodes, dead CHs, network lifetime and energy consumption. This section compares the simulation result of the proposed IGWO algorithm with traditional GWO, HSCA [8] and GALBCA [10].

#### 5.2.1. Comparison of Dead CHs

The number of active states of cluster heads (CHs) has a significant impact on the WSN’s ability to operate reliably. This enables utilities to quickly and effectively communicate customer information and take the necessary action.

The results shown in Table 3 show that the number of CHs goes dead in different rounds (i.e., 500, 1000, 1500 and 2000) for the IGWO, traditional GWO, HSCA, and the GALBCA. Here, it can be observed that the number of CHs dies in IGWO is a minimum compared to any other approach. In 500 rounds, there are merely three dead CHs in IGWO, whereas there are four, five, and seven in other approaches. Similarly, for 1000, 1500, and 2000 rounds, the HSCA has performed better after IGWO. In the HSCA approach, the number of CHs that die in 1000 rounds is 12, while in 1500 rounds, it is 19, and in 2000 rounds, it is 33, which are less than other approaches except for IGWO. However, for the same rounds, the number of dead CHs is found to be 11, 17 and 31 in case of IGWO. Figure 10 shows the performance of IGWO in terms of dead CHs per round as compared to the traditional GWO, HSCA and GA-based approaches for the same objectives.

#### 5.2.2. Comparison of Dead Sensor Node

The capacity of the sensor to exchange information with CH is further enhanced by the clustering, which enables a change in network topology. On the other hand, a sensor that is distant from their CH will use more energy and have a higher chance of becoming dead. As a result, it might not give customers the information they need at the appropriate moment to take critical action.

Table 4 shows the results for dead sensor nodes in the different rounds. In this analysis, the number of dead sensor nodes is shown cumulatively for 500, 1000, 1500 and 2000 rounds. The test results show that in the IGWO approach, the dead sensor nodes are 17, 51, 149 and 279 in 500, 1000, 1500 and 2000 rounds, respectively. In contrast to other techniques, with the exception of HS, the increase in dead sensors is linear in the suggested methodology. As a result, the increased number of rounds makes the proposed strategy more successful in terms of dead sensors.

Figure 11 displays the variation in dead sensors for various methodologies, such as improved harmony search, for each round. The proposed IGWO-based load-balanced technique outperforms all alternatives in this scenario by a large margin.

#### 5.2.3. Comparison of Energy Consumption

The network lifetime in real life is based on the energy lost during information exchange. As a result, the suggested clustering aims to maximize WSN lifetime by lowering energy usage. The energy usage is influenced by the proximity of the sensors to their CH, as explained in Section 5.2.2. However, because there is no information transmission, the energy usage may decrease as the number of dead sensors increases. Although this is obviously undesired, it will result in a loss of customer connectivity with the utilities. Maintaining sensors while lowering energy usage through the energy-efficient clustering of WSNs is a significant problem in this situation.

Table 5 compares energy consumption in different approaches in 500, 1000, 1500 and 2000 rounds. The energy consumption in IGWO is 202, 389, 581 and 698 in 500, 1000, 1500 and 2000 rounds, respectively. The results show less energy consumption in IGWO and the maximum is in GALBCA [10].

Figure 12 shows the energy consumption in each round, and here, it can be observed that the curve for IGWO is always below the others due to the improvement in the hunting mechanism of gray wolves and their social hierarchy.

Table 6 displays the energy usage fluctuation from 500 rounds to 2000 rounds. In Figure 13, the energy consumption is represented in terms of the number of dead sensor nodes from 500 to 2000 rounds. The rate of dead sensors is high here as well in the first 500 rounds, but it goes down as the number of increment rounds increases.

#### 5.2.4. Comparison of Lifetime

The CHs’ active status determines the WSNs’ lifetime; therefore, keeping them alive for as long as possible is essential for effective communication. Table 7’s findings demonstrate that the proposed IGWO technique consistently results in fewer CH deaths, with 3, 8, 6 and 14 CHs dying in each of the 500 rounds. It is evident that the HSCA strategy performs almost equal to the proposed approach.

Figure 14 compares network lifetime in terms of the number of CHs die in different rounds. Here, it can be observed that in 500, 1500 and 2000 rounds, the minimum number of CHs die in the case of the IGWO, whereas the maximum CHs die in the case of the GA-based approach. In contrast, fewer dead CHs are seen in the GA-based strategy in rounds 501–1000. When compared to other approaches, the IGWO and HSA are determined to have the fewest dead CHs as the number of rounds grows.

The dead sensor nodes need to be evaluated based upon the first and last dead sensor for assessing the network lifetime. Table 8 compares the different approaches for this analysis. Here, the first node dies in GALBCA in 291, HSCA in 407, traditional GWO in 430 and IGWO in 496 rounds. Therefore, the network failure occurs in GALBCA at an early stage as compared with other approaches. However, the last node that dies in the genetic algorithm is in 1922 rounds, and at this stage, in this approach, most of the nodes were dead, i.e., 454 nodes are dead out of 500, as shown in column 4 of Table 8.

Table 9 shows the number of sensor nodes die at the different intervals in a range of 0–2000 rounds. Only 17, 34, 98 and 130 sensor nodes die between different steps of rounds in the IGWO approach, whereas a large difference between the dead sensors occurs in other approaches, except HSCA. Figure 15 represents the network lifetime in context of the number of sensors die. From Figure 15, it can be observed that the dead sensors are minimum at the end of 500 and 1000 in the IGWO and in 1500 rounds, GA offers the minimum, whereas in 2000 rounds, the dead sensors are again maximum in a GA-based approach.

## 6. Conclusions

This work presented the energy-efficient load-balanced clustering for sustainable WSNs using improved gray wolf optimization (IGWO). For sustainable WSNs, a performance matrix has been developed based on the dead sensors, dead CHs, energy consumption, lifetime, and the algorithm’s convergence rate. The performance of various computing techniques for the energy-efficient load-balanced clustering of WSNs is also presented compared to the proposed approach. In the proposed work, the traditional GWO has improved, and its computation tendency is explored for the energy-efficient clustering of WSNs in a smart city environment. The improvement in the performance matrix allows better communication among sensors, and it will provide extensive area coverage for customer satisfaction. Therefore, in a smart city environment, the increase in lifetime and reduction in energy consumption improves the reliability of operation. Moreover, for robustness, a comparative analysis is demonstrated, and it is observed that IGWO performed better than the existing computing approaches. The reliability of WSN can be improved for sustainable cities and hence the customers’ satisfaction can be improved by serving better wirelessly from a remote place and for a long time.

## Figures and Tables

**Figure 1 sensors-22-07113-f001:**
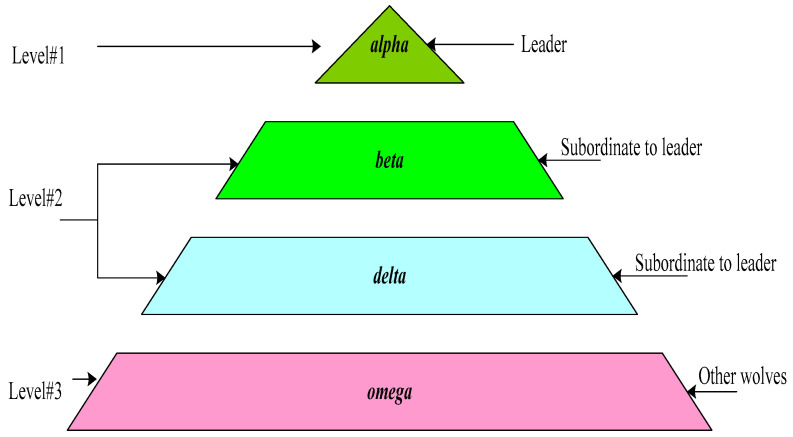
Social hierarchy of wolves.

**Figure 2 sensors-22-07113-f002:**
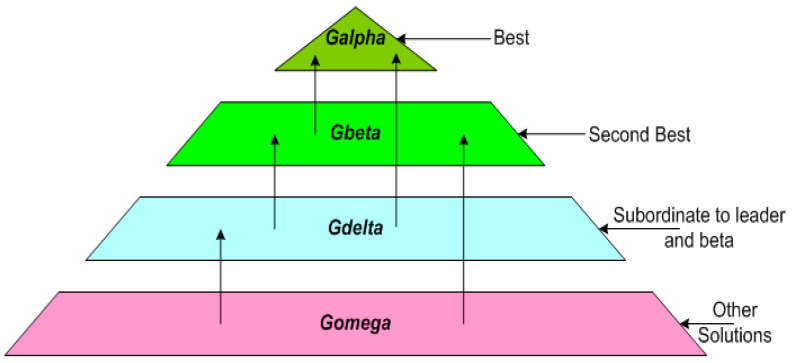
Social hierarchy in the proposed algorithm.

**Figure 3 sensors-22-07113-f003:**
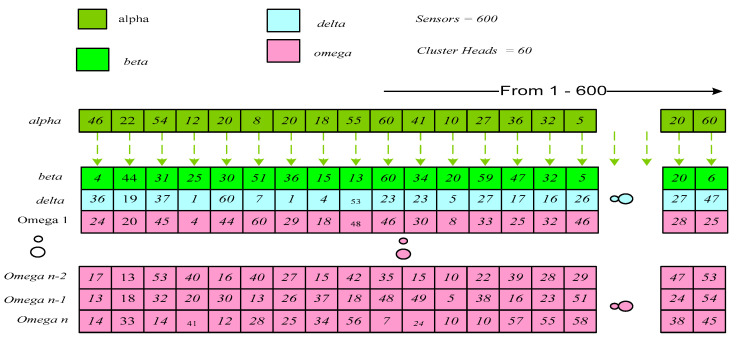
Initialization.

**Figure 4 sensors-22-07113-f004:**
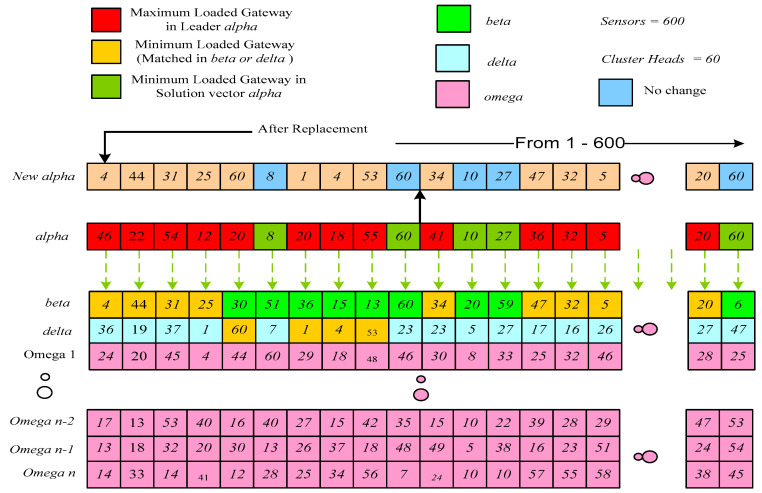
Hunting (illustration of updating *alpha*).

**Figure 5 sensors-22-07113-f005:**
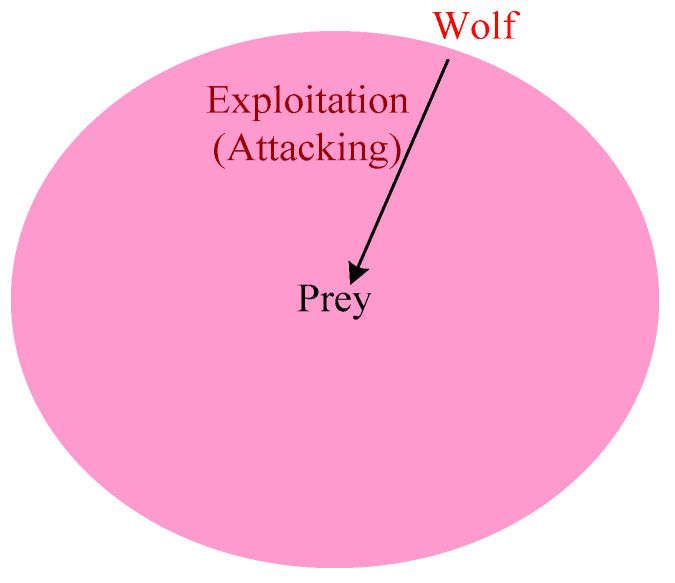
Exploitation.

**Figure 6 sensors-22-07113-f006:**
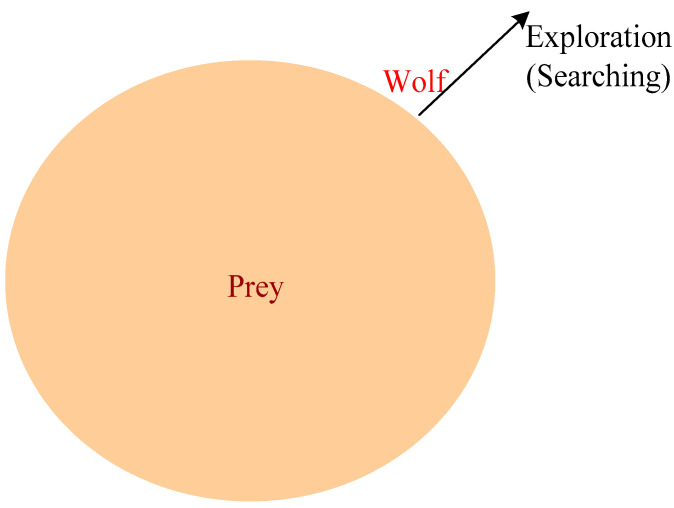
Exploration.

**Figure 7 sensors-22-07113-f007:**
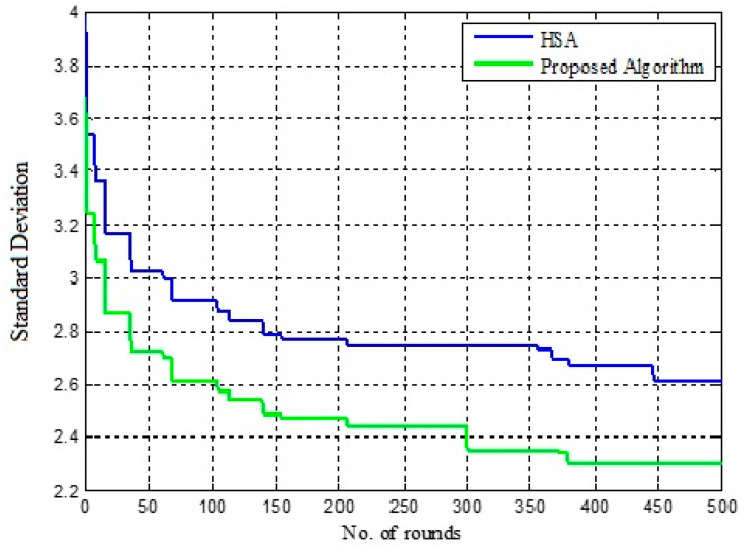
Fitness improvement in 500 rounds.

**Figure 8 sensors-22-07113-f008:**
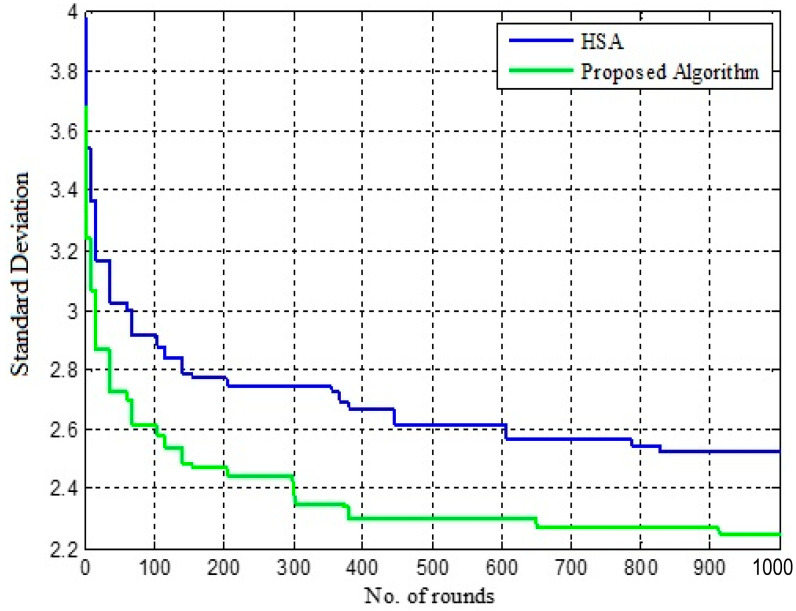
Fitness improvement in 1000 rounds.

**Figure 9 sensors-22-07113-f009:**
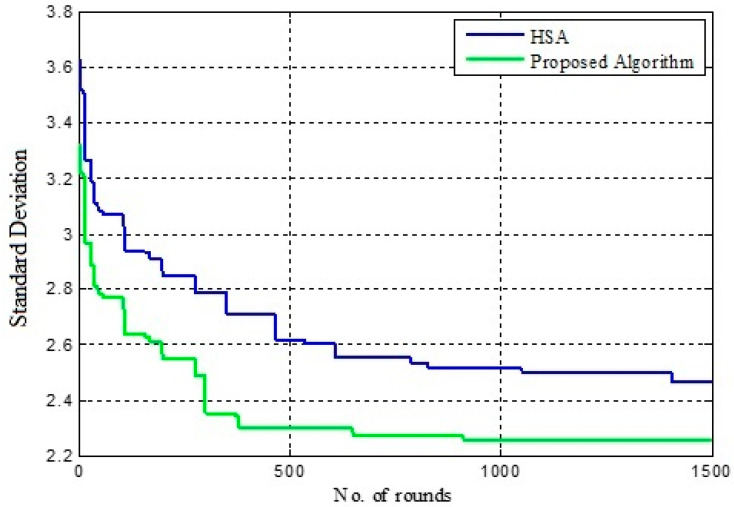
Fitness improvement in 1500 rounds.

**Figure 10 sensors-22-07113-f010:**
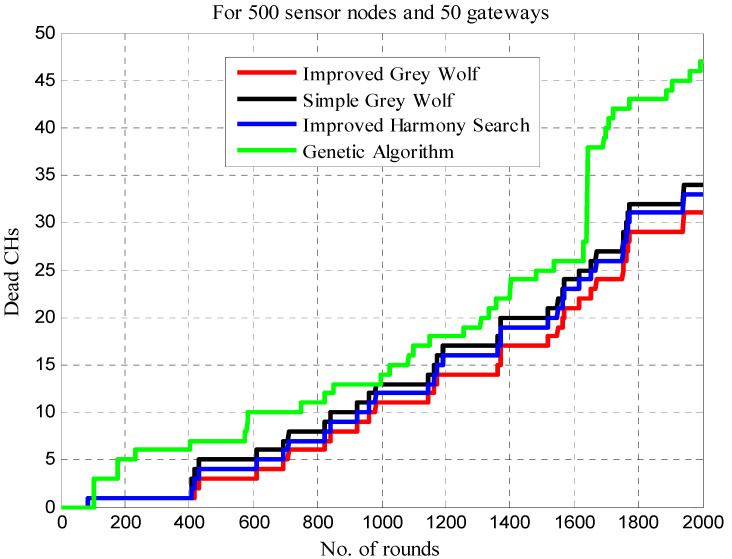
Dead CHs vs. number of rounds.

**Figure 11 sensors-22-07113-f011:**
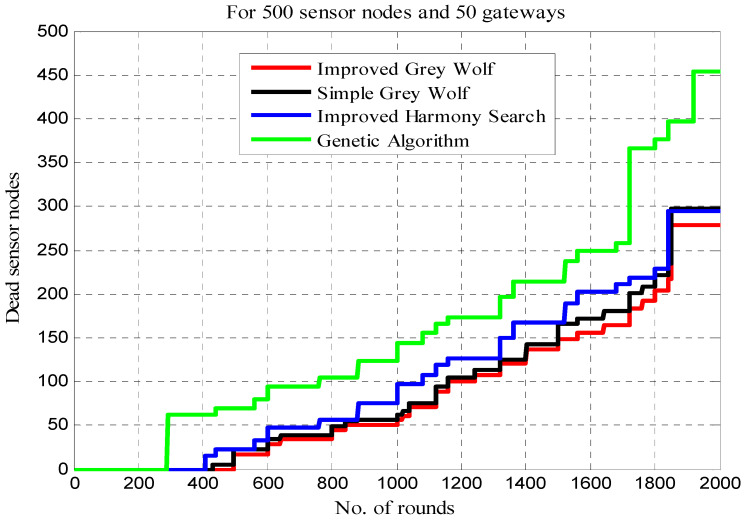
Dead sensor nodes vs. number of rounds.

**Figure 12 sensors-22-07113-f012:**
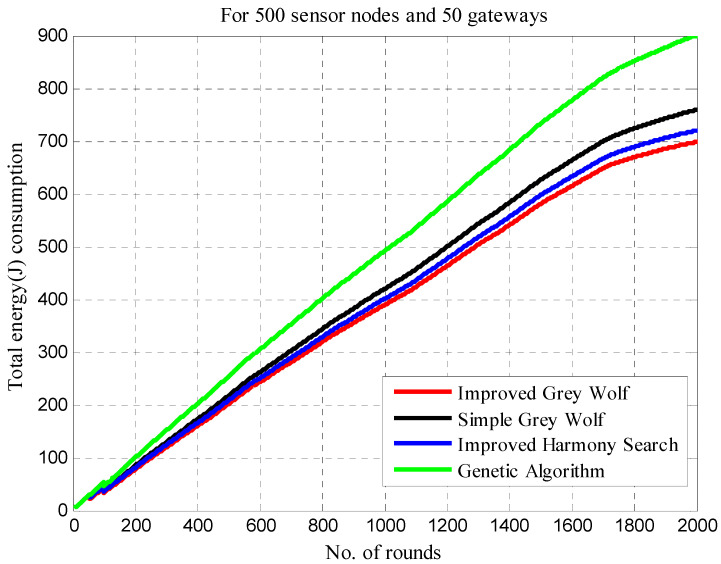
Total energy consumption.

**Figure 13 sensors-22-07113-f013:**
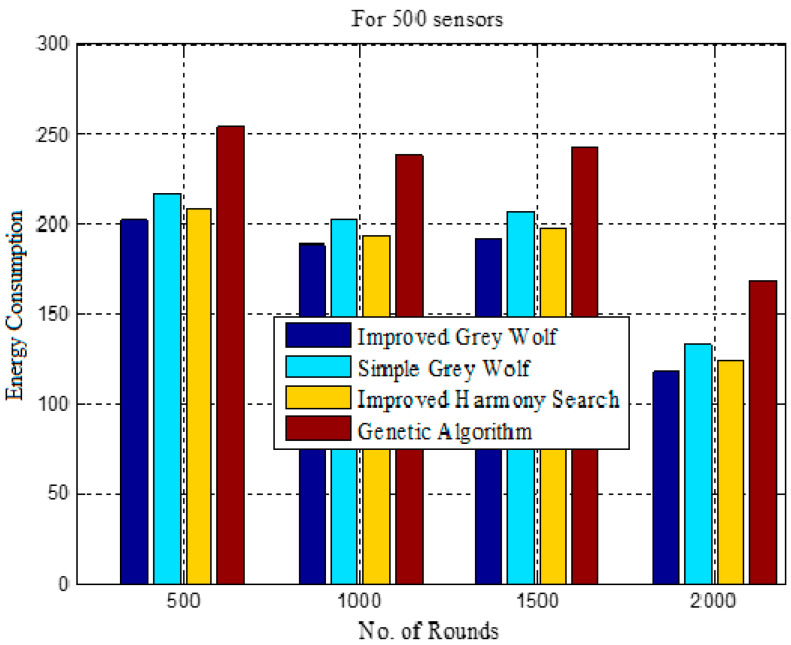
Energy consumption in different number of rounds.

**Figure 14 sensors-22-07113-f014:**
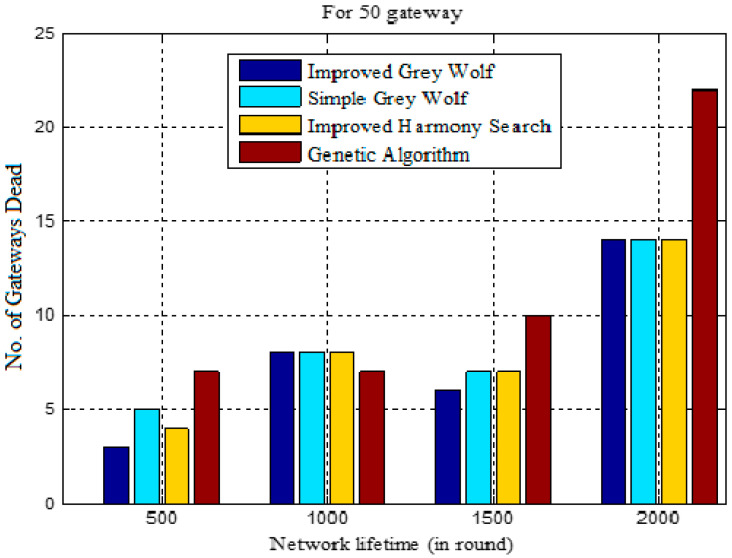
Network lifetime (with CHs) in different number of rounds.

**Figure 15 sensors-22-07113-f015:**
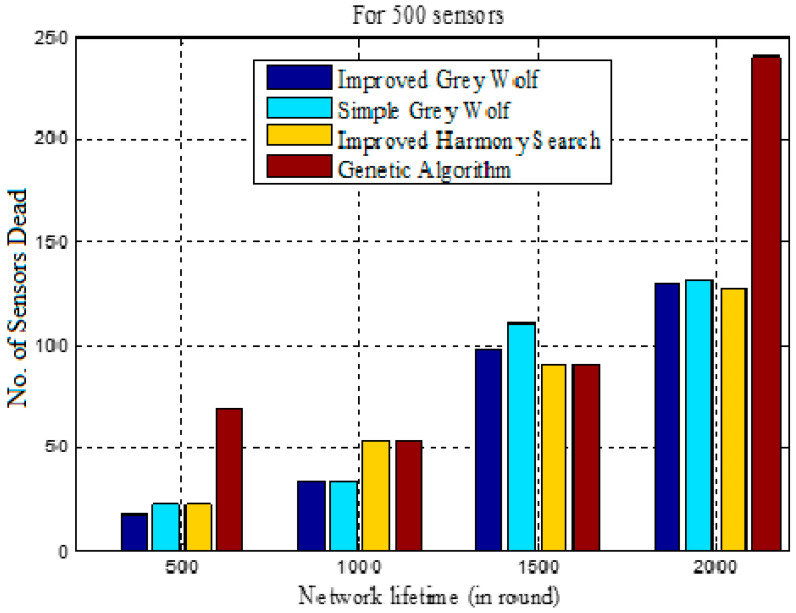
Network lifetime (with sensors).

**Table 1 sensors-22-07113-t001:** Description of the parameters used.

S.No.	Parameters	Values
1	Round	2000
2	Sink location	500 × 250
3	*d* _0_	60
4	CH communication range	130 m
5	*d_max_*	65
6	Sensor initial energy	2.0 J
7	CH initial energy	12 J
8	Packet size	4000 bits
9	Message Size	500 bits
10	Area (m^2^)	500 × 500
11	Sensor	500
12	CH	50

**Table 2 sensors-22-07113-t002:** Variation in standard deviation for IGWO and HSCA.

Rounds	Algorithm	Standard Deviation
500	Proposed IGWO	2.3016
	HSCA	2.6145
1000	Proposed IGWO	2.2516
	HSCA	2.5245
1500	Proposed IGWO	2.2516
	HSCA	2.4651

**Table 3 sensors-22-07113-t003:** Comparison of dead CHs in different rounds.

Clustering Techniques	Dead CHs in Rounds			
	500	1000	1500	2000
IGWO	3	11	17	31
Traditional GWO	5	13	20	34
HSCA	4	12	19	33
GA	7	14	25	47

**Table 4 sensors-22-07113-t004:** Comparison of dead sensor nodes.

Clustering Techniques	Dead Sensors in Rounds			
	500	1000	1500	2000
IGWO	17	51	149	279
Traditional GWO	22	56	166	297
HSCA	22	76	167	295
GA	69	123	214	454

**Table 5 sensors-22-07113-t005:** Energy consumption in different rounds.

Clustering Techniques	Energy Consumption in Different Rounds			
	500	1000	1500	2000
IGWO	202	389	581	698
Traditional GWO	217	420	626	758
HSCA	208	401	597	719
GA	254	491	733	900

**Table 6 sensors-22-07113-t006:** Comparison of energy consumption in different approaches.

Clustering Techniques	Energy Consumption in Rounds			
	0–500	501–1000	1001–1500	1501–2000
IGWO	202	188	192	118
Traditional GWO	217	203	207	133
HSCA	208	193	197	123
GA	254	238	242	168

**Table 7 sensors-22-07113-t007:** Comparison of dead CHs for network lifetime.

Clustering Techniques	Lifetime in Rounds			
	0–500	501–1000	1001–1500	1501–2000
IGWO	3	8	6	14
Traditional GWO	5	8	7	14
HSCA	4	8	7	14
GA	7	7	11	22

**Table 8 sensors-22-07113-t008:** Comparison of first and last sensor nodes die in different techniques.

Clustering Techniques	First Nodes Die in No. of Rounds	Last Nodes Die in No. of Rounds	Total Nodes Die in 2000 rounds
IGWO	496	1848	279
Traditional GWO	430	1849	297
HSCA	407	1836	295
GA	291	1922	454

**Table 9 sensors-22-07113-t009:** Comparison of dead sensor nodes for network lifetime.

Clustering Techniques	Lifetime in rounds			
	0–500	501–1000	1001–1500	1501–2000
IGWO	17	34	98	130
Traditional GWO	22	34	110	131
HSCA	22	54	91	128
GA	69	54	91	240

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
