# Peer review of "GWLBC: Gray Wolf Optimization Based Load Balanced Clustering for Sustainable WSNs in Smart City Environment"

_sensors, 2022, doi:10.3390/s22197113_

Round 1
Reviewer 1 Report
This is a very interesting research but will improved using a more specfic case of study iincluding more historical data.
Is very important an analyze very descriptive with multivariable análisis to this research to improve this scenario.
In addition, is very important make a comparative with related research in a real world problems of Industry 4.0.
Many situations associated in this moment with similar models including the effect of human factor.
A correct design of experiments is very important to understand the possible variability in these methodological process.
Is very important describe a better comparative with another similar implementatios as in:
Jian Liu, Sanming Liu:
An Improved Dual Grey Wolf Optimization Algorithm for Unit Commitment Problem. LSMS/ICSEE (2) 2017: 156-163.
Author Response
Response to the reviewer’s comments
Reviewer#1:
Comment#1: This is a very interesting research but will improved using a more specific case of study including more historical data.
Response Author are thankful to the renowned reviewer for their appreciation and valuable observations. In the proposed work, a large network is considered which has 500 nodes in the area of 500m2.
In this work, we have compared the proposed approach with the various existing approaches.
Where, the existing work reveals that there is no such approach exits in the proposed scenario which considers the historical data because the clustering of wireless sensor nodes deployed in random manner which has been solved with different computational techniques. In wireless ad-hoc networks the major problem is to create clusters of sensor nodes. A typical clustering algorithm helps us to find a cluster head which can pass sensors data to the base station. These algorithms are implemented in a simulation environment.
However, our approach can also be applied over a historical data, which will become a different approach for data analysis rather than WSN Clustering and Routing Algorithm in simulation environment. Therefore, the renowned reviewer’s suggestion will definitely be considered in our future work where we shall apply different machine learning approaches over a historical data.
Comment#2: Is very important an analyse very descriptive with multivariable analysis to this research to improve this scenario.
Response: Authors are afraid to say that we did not find any multivariable data which can be used in the proposed scenario. Because it is a different approach rather than like data analysis using machine learning approaches such as K-Means Clustering etc. However, as per the understanding here author would like to convey that the proposed approach is applicable for the multi objective functions where several parameters are evaluated. Therefore, optimization is performed on assigning the different weightage to various component, over the other.
Authors are thankful to the renowned reviewer for his suggestion which shall definitely be considered in our future work, where we shall apply different machine learning classification techniques over a historical data with multiple features.
Comment#3: In addition, is very important make a comparative with related research in a real-world problems of Industry 4.0.
Response: Thank you for your valuable observation. The suggested related work has been added in the text.
As per the suggestion, it has been realized that the Researchers, manufacturers, and application developers have recently shown a lot of interest in the development of new digital industrial technology known as Industry 4.0. Industry 4.0 aims to connect and integrate all elements of the conventional manufacturing environment in order to facilitate quicker, more adaptable, and more efficient production processes that result in higher-quality products at lower costs.
However, it has been observed from the relevant literature that an enhanced GWO technique is not available for effective load balanced clustering in Industry 4.0. Whereas, proposed work investigates improved GWO for energy-efficient load balanced clustering.
Comment#4: Many situations associated in this moment with similar models including the effect of human factor.
Response: In response to this critical observation author would like to say that the proposed approach is implemented for the sensor deployment in a specified area where, energy consumption due to data communication depends upon the proximity of sensors from the cluster head. Therefore, in this scenario, the involvement of human factor may turn the approach of energy efficient clustering in different aspect which is not in the scope of the proposed work. However, this is very innovative idea which can be addressed in extension of this work in future.
Comment#5: A correct design of experiments is very important to understand the possible variability in these methodological process.
Response: Thank you for the valuable observation and suggestions. In the proposed work, the performance of several approaches have been evaluated with same network and under similar operating conditions. Here, GA, PSO, HSA and GWO (in conventional form) and the improved GWO (proposed) have been tested on a 500 node system in terms of the number of dead sensors, dead clusters heads and energy savings. Though, in some cases the results of the existing approaches are found convenience whereas, when same approach is evaluated for other parameters, it yields different results. However, the variations in the results may vary different across these approaches but it will affect the computational time if operating constraints are managed differently. However, the optimal results and computational time of these approaches is highly dependent on the operator experience and operating constraints. In this scenario, the pareto-optimal results are obtained which are comparable to existing approaches.
Comment#6: Is very important describe a better comparative with another similar implementations as in: Jian Liu, Sanming Liu: An Improved Dual Grey Wolf Optimization Algorithm for Unit Commitment Problem. LSMS/ICSEE (2) 2017: 156-163.
Response: Authors are thankful to the renowned reviewer for critical observation. Here, we would like to submit that the application of the GWO in optimization has wide scope in engineering problems. However, the formulation of optimization function may vary significantly due to imposition of the operating constraint for specific problems. In this scenario, the optimization performed which may not be equally applicable for different type of problems. The suggested reference has been added in the text.

Reviewer 2 Report
Dear Authors,
The paper is well written and structured, however I have some remarks, which are listed below. Some errors, which I found are marked in the annotated PDF file, which is attached to the review:
- in equation 1 '}' is missing
- upgrade the quality of Fig 2 - it's worse than the other ones
- in Fig 3 pseudocode there are Galpha and G_alpha are they different or is it some kind of a mistake?
- The pseudocodes could be formatted in a better manner and additionally described in the paper text. Right now, the chosen template makes them hard to follow.
- in all the tests and descriptions you write about 500 sensors. However in Fig 5 it seems that you have 600 instead. The same goes for population matrix shape and subsequent algorithm. Could you elaborate on that or introduce appropriate changes?
- The results section could be upgraded as it is quite repetetive, especially the energy related section could be shortened - it is quite obvious that the more power the devices consume the more it is possible that they will die out. Without any specific information on the device, the employed standard etc. the number of dead devices do not tell much.

Author Response
Reviewer#2:
Dear Authors,
The paper is well written and structured; however, I have some remarks, which are listed below. Some errors, which I found are marked in the annotated PDF file, which is attached to the review:
Response: Authors are very thankful to the renowned reviewer for their appreciation and valuable comments. The revised manuscript has been updated to reflect the suggested modifications from the marked-up, annotated pdf.
Comment#1: in equation 1 '}' is missing
Response: The error pointed out by the reviewer has been corrected in the revised manuscript.
Comment#2: upgrade the quality of Fig 2 - it's worse than the other ones
Response: In the revised version the quality of Fig. 2 has been improved.
Comment#3: in Fig 3 pseudocode there are Galpha and G_alpha are they different or is it some kind of a mistake?
Response: The mistake pointed out by the reviewer has been corrected in the revised manuscript, which is as follows:
Step7.1.2.1.1: Set G_Load1=Galpha[i].
Step7.1.2.1.2: Set Galpha[i] = Gbeta[i].
Set 7.1.2.1.3: Set Gbeta[i]=G_Load1
Comment#4: The pseudocodes could be formatted in a better manner and additionally described in the paper text. Right now, the chosen template makes them hard to follow.
Response: The pseudocode has been improved and is now presented in full for easier comprehension, as per the suggestion.
Comment#5: in all the tests and descriptions you write about 500 sensors. However, in Fig 5 it seems that you have 600 instead. The same goes for population matrix shape and subsequent algorithm. Could you elaborate on that or introduce appropriate changes?
Response: We have used 600 sensors for illustration in Fig 5 and Fig 6. Whereas, in all test system result analysis, we have used 500 sensors. So, there is no change.
Comment#6: The results section could be upgraded as it is quite repetetive, especially the energy related section could be shortened - it is quite obvious that the more power the devices consume the more it is possible that they will die out. Without any specific information on the device, the employed standard etc. the number of dead devices do not tell much.
Response: According to the respected reviewer's advice, the results section has been upgraded.

Round 2
Reviewer 2 Report
Dear Authors,
Thank you for your response and introducing the suggested changes. I recommend the paper to be published.